# Overexpression of *AtMYB2* Promotes Tolerance to Salt Stress and Accumulations of Tanshinones and Phenolic Acid in *Salvia miltiorrhiza*

**DOI:** 10.3390/ijms25074111

**Published:** 2024-04-08

**Authors:** Tianyu Li, Shuangshuang Zhang, Yidan Li, Lipeng Zhang, Wenqin Song, Chengbin Chen

**Affiliations:** College of Life Sciences, Nankai University, Tianjin 300071, China; litianyu582@163.com (T.L.); ylnmll99@163.com (S.Z.); danicali2021@163.com (Y.L.); nknanhai@163.com (L.Z.); songwq@nankai.edu.cn (W.S.)

**Keywords:** *AtMYB2*, *Salvia miltiorrhiza*, salt tolerance, tanshinones, phenolic acid

## Abstract

*Salvia miltiorrhiza* is a prized traditional Chinese medicinal plant species. Its red storage roots are primarily used for the treatment of cardiovascular and cerebrovascular diseases. In this study, a transcription factor gene *AtMYB2* was cloned and introduced into *Salvia miltiorrhiza* for ectopic expression. Overexpression of *AtMYB2* enhanced salt stress resistance in *S. miltiorrhiza*, leading to a more resilient phenotype in transgenic plants exposed to high-salinity conditions. Physiological experiments have revealed that overexpression of *AtMYB2* can decrease the accumulation of reactive oxygen species (ROS) during salt stress, boost the activity of antioxidant enzymes, and mitigate oxidative damage to cell membranes. In addition, overexpression of *AtMYB2* promotes the synthesis of tanshinones and phenolic acids by upregulating the expression of biosynthetic pathway genes, resulting in increased levels of these secondary metabolites. In summary, our findings demonstrate that *AtMYB2* not only enhances plant tolerance to salt stress, but also increases the accumulation of secondary metabolites in *S. miltiorrhiza*. Our study lays a solid foundation for uncovering the molecular mechanisms governed by *AtMYB2* and holds significant implications for the molecular breeding of high-quality *S. miltiorrhiza* varieties.

## 1. Introduction

Abiotic stress, such as drought, salinity, and extreme temperatures, has a detrimental impact on plant growth, development, and productivity [1]. Salt stress limits plant growth by disturbing cellular biochemistry and physiology through the excessive production of reactive oxygen species (ROS). The primary forms of ROS include singlet oxygen (^1^O_2_), hydrogen peroxide (H_2_O_2_), and superoxide radical (O_2_^−^) [2]. Oxidative stress is a secondary stress triggered by salt stress in plants, where salt stress promptly leads to the buildup of harmful ROS and subsequent oxidative damage [3]. Plants have evolved diverse mechanisms to mitigate oxidative stress, including the synthesis of antioxidants and the activation of stress response pathways [4]. Many enzymatic and non-enzymatic antioxidant scavengers help to maintain ROS at suitable levels to avoid toxicity in plants under salt stress. The scavengers of ROS include superoxide dismutase (SOD), peroxidase (POD), and catalase (CAT), among others. These enzymes are activated following stress, thereby aiding plants in better coping with adverse environmental conditions [5,6].

Transcription factors (TFs) are specialized proteins that serve as essential regulators in the precise control of gene expression by binding to specific promoter regions [7]. The MYB family is one of the largest TF families in plants, and many MYB family members have been found to be involved in coping with salt stress in plants. For instance, *SiMYB16* can enhance plant salt tolerance by regulating the biosynthesis of lignin and suberin [8]. The overexpression of *GmMYB81* markedly increases seed germination rates and promotes the growth of green seedlings under conditions of salt and drought stress. This suggests that *GmMYB81* has the potential to enhance plant tolerance to salt and drought stress, specifically during seed germination [9]. The overexpression of *AtMYB20* in *Arabidopsis* downregulated the expression of *PP2Cs*, the negative regulators of ABA signaling which encode type 2C serine/threonine protein phosphatases, to help plants adapt to higher salt concentrations [10]. Research has demonstrated that SlMYB102 is involved in scavenging ROS by regulating the activities of enzymatic scavengers and the accumulation of antioxidants in response to salt stress [11].

Salvia miltiorrhiza Bunge, belonging to the Lamiaceae family, is a valuable traditional Chinese medicinal plant species commonly known as Danshen. Bioactive components of Danshen have the effects of anti-oxidation, anti-inflammation, and improving cardiovascular and cerebrovascular diseases. For example, tanshinone IIA can contribute to the treatment of coronary atherosclerotic heart disease (CHD) by reducing myocardial I/R injury. Salvianolic acid B can play a vascular relaxation role in vascular smooth muscle cells by inhibiting Ca^2+^ influx [12,13]. There are two major groups of active pharmaceutical ingredients in Danshen: lipophilic tanshinones (tanshinone I, T-I; tanshinone IIA, T-IIA; cryptotanshinone, CT) and hydrophilic phenolic acids (salvianolic acid B, SalB; rosmarinic acid, RA; caffeic acid, CA) [14]. Given the rising clinical demand for *Salvia miltiorrhiza*, it is imperative to employ innovative metabolic engineering strategies to enhance and improve the biosynthesis of bioactive compounds in this plant. Recently, several MYB transcription factors (TFs) have been identified as regulators of tanshinones and phenolic acid biosynthesis in *Salvia miltiorrhiza*. Typically, overexpression of both *SmMYB9b* [15] and *SmMYB98b* [16] can promote tanshinone concentration in the hairy roots of *Salvia miltiorrhiza*, while *SmMYB111* [17] and *SmMYB2* [18] are both positive regulator of phenolic acid biosynthesis. Studies have shown that some MYB family genes of *Arabidopsis* can also promote increases in salvianolic acid content in *S. miltiorrhiza*. *AtMYB75* (also known as *AtPAP1*) enhances phenolic acid biosynthesis by upregulating the expression of *SmbHLH51* in *Salvia miltiorrhiza* [19], and *AtMYB12* facilitates the accumulation of SalB by enhancing the transcriptional levels of genes participating in the biosynthetic pathways [20].

Previous research has shown that the expression of *AtMYB2* is induced under conditions of water stress, high salinity, and exposure to abscisic acid (ABA) [21]. Also, transcription of *AtMYB2* could be induced in response to Pi deficiency [22] and low-oxygen conditions [23]. Furthermore, it has been reported that *AtMYB2* is related to the synthesis of cytokinin, and therefore affects the senescence of leaves [24] and the whole plant [25]. In the present study, we found that *AtMYB2* could not only improve the salt tolerance of transgenic *S. miltiorrhiza*, but also promote the accumulation of tanshinones and phenolic acid in both hairy roots and plants.

## 2. Results

### 2.1. Generation and Molecular Characterization of Transgenic Hairy Roots and Plants

The plasmid *pCAMBIA1301-AtMYB2* was constructed with the gene regulated by the CaMV35S promoter to investigate the role of *AtMYB2* in transgenic *Salvia miltiorrhiza* (Appendix A). Following Agrobacterium-mediated transformation, genomic DNA was extracted from transgenic hairy roots, transgenic plants, and the control. PCR results are shown in Appendix A. The expression profiles of *AtMYB2* were analyzed by qRT-PCR (Figure 1). Three transgenic hairy root lines (Lines 4, 5, and 6) and three transgenic plant lines (Lines 2, 5, and 7) with relatively higher transcript accumulation were selected for further investigation. Molecular and quantitative experiments demonstrated that the exogenous *AtMYB2* was stably integrated into the genomes of *S. miltiorrhiza* hairy roots and plants, albeit with varying expression levels.

### 2.2. AtMYB2 Promotes Tolerance to Salt Stress in S. miltiorrhiza

To explore the correlation between *AtMYB2* expression and salt tolerance in *S. miltiorrhiza*, we conducted short-term and long-term salt stress experiments utilizing both detached leaves and whole seedlings. After four days of treatment with 100 mM and 250 mM salt solutions, the detached leaves of three transgenic lines exhibited a greener appearance compared to both the wild type (WT) and the empty vector control pCAMBIA1301, showcasing an augmented level of salt tolerance (Appendix A). In the long-term salt stress experiments, WT and transgenic plantlets were subjected to 15 d of salt stress. At 3 days after treatment (3 DAT), the leaves of wild-type (WT) and pCAMBIA1301 plants displayed yellowing, while the transgenic lines showed no discernible changes. At 7 DAT, the leaves of WT and pCAMBIA1301 plants displayed severe damage and necrosis, whereas the transgenic lines maintained a comparatively greener appearance with reduced wilting, although some leaf damage was evident. However, with the prolonged duration of salt treatment, all tested *S. miltiorrhiza* seedlings perished at 15 DAT. In the control group receiving regular watering, all *S. miltiorrhiza* seedlings exhibited robust and uniform growth (Figure 2). The above results indicate that *AtMYB2* transgenic *S. miltiorrhiza* seedlings exhibit higher resistance to salt stress compared to both WT and pCAMBIA1301 seedlings.

### 2.3. AtMYB2 Reduces ROS Accumulation under Salt Stress

Under salt stress, plants often experience increased production of ROS. To assess whether genetic modifications could alleviate ROS accumulation under salt stress, nitroblue tetrazolium (NBT) and diaminobenzidine (DAB) staining were employed to visualize the accumulation of O^2−^ and H_2_O_2_, respectively. The results revealed minimal disparity among the wild-type, pCAMBIA1301, and transgenic *S. miltiorrhiza* under normal conditions. However, upon treatment with 250 mM NaCl, transgenic *S. miltiorrhiza* leaves exhibited reduced staining intensities compared to the WT, indicating a significant reduction in salt-induced ROS production due to the overexpression of *AtMYB2* (Figure 3A,B). Further quantification of O^2−^ and H_2_O_2_ content corroborated these observations, revealing a notable decrease of 60.4–68.4% in O^2−^ and a 32.4–39.5% decrease in H_2_O_2_ accumulation in transgenic lines compared to the WT (Figure 3C,D)

### 2.4. AtMYB2 Improves Antioxidant Capacity under Salt Stress

Electrolyte leakage (EL) and total chlorophyll content were measured after 7 days of salt stress treatment. In comparison to the WT and pCAMBIA1301 lines, all transgenic *AtMYB2* lines exhibited significantly lower EL, with the OE-AtMYB2-L2 line showing a reduction of one third compared to the WT and pCAMBIA1301 lines (Figure 4A). Furthermore, the transgenic *AtMYB2* lines showed higher total chlorophyll content than that in the WT, with the *OE-AtMYB2-L2* line showing the highest value (Figure 4B). Furthermore, there was a decrease in MDA levels across all transgenic lines compared to the WT, indicating a mitigation of cell membrane damage in the *AtMYB2* transgenic plants (Figure 4C).

Given that the accumulation of ROS was reduced in *AtMYB2* transgenic plants under salt stress conditions, we wondered whether the activities of enzymatic ROS scavengers were affected. We noticed that the enzymatic activities of SOD, POD, and CAT were significantly elevated in all *AtMYB2* transgenic lines compared to the WT, with increases ranging from 29.7% to 74.2% (Figure 4D–F). The data indicated that under salt stress conditions, the enzymatic scavenging activities were significantly enhanced in the transgenic lines, and *AtMYB2* improved the ability to resist salt stress by enhancing the antioxidant capacity of transgenic *S. miltiorrhiza* under salt stress.

### 2.5. AtMYB2 Promotes the Accumulation of Tanshinones and Salvianolic Acid

Compared to the wild type, the overexpression of *AtMYB2* led to alterations in the secondary metabolite content of transgenic *S. miltiorrhiza* hairy roots and plants. The *AtMYB2* transgenic hairy roots appeared much redder and thicker than those of the control plants (Figure 5A). In transgenic hairy roots, the content of Sal B was highest in *OE-AtMYB2-L4*, reaching 20.41 mg/g dw, which was 22.7% higher compared with the WT. Upon observation, it was noted that the tanshinone extracts from the *AtMYB2* transgenic hairy root lines exhibited a deeper red color compared with the control lines (Figure 5D), suggesting a potential elevation in tanshinone concentration. Through high-performance liquid chromatography (HPLC) detection, the transgenic hairy root line with the highest content of each tanshinone was *OE-AtMYB2-L6*, which increased by 27.1% (TI), 24.6% (TIIA), and 20.6% (TT) compared with the WT, respectively. The content of CT in each transgenic hairy root line showed no significant differences (Figure 5G).

As for the roots of plants, phenotypically, the roots of transgenic plants exhibited fewer fine branchings and a deeper red coloration (Figure 5B). The transgenic plant lines produced much higher levels of Sal B and total tanshinone than the WT. Meanwhile, the Sal B and total tanshinone content increased by 38.5% (40.5 mg/g dw) and 20.5% (11.29 mg/g dw) in *OE-AtMYB2-L5* when compared with those in the WT, respectively. Similarly, compared to the control group, the tanshinone extract from transgenic *S. miltiorrhiza* roots also exhibited a deeper red color. However, *AtMYB2* transgenic plant lines had more of an effect on the production of TI and TIIA than CT (Figure 5H).

### 2.6. AtMYB2 Promotes Tanshinones and Salvianolic Acid Content by Upregulating the Expression of Biosynthesis Genes

To elucidate the mechanisms underlying these changes in the contents of salvianolic acid and tanshinones, we first examined the transcript levels of eight key enzyme genes in the salvianolic acid biosynthesis pathway and eight enzyme genes in the tanshinone biosynthesis pathway in transgenic hairy roots. The results indicated that, compared to the wild type, the expression levels of *SmPAL1*, *SmC4H2*, *SmTAT1*, and *SmHPPR1* in the salvianolic acid biosynthesis pathway were significantly increased (ranging from 1.30 to 4.15 times higher than the wild type) in transgenic hairy roots (*p* < 0.05), while the expression levels of the remaining four genes showed no significant differences. The expression levels of all eight key enzyme genes involved in the tanshinone biosynthetic pathway were significantly upregulated, ranging from 1.42- to 6.69-fold higher in the transgenic hairy root lines compared to the wild type (Appendix A).

Subsequently, we selected two *AtMYB2* transgenic *S. miltiorrhiza* lines, Line5 and Line7, with higher salvianolic acid and tanshinones content, and conducted a more detailed expression analysis of the key enzyme genes in both the salvianolic acid and tanshinone biosynthesis pathways. The results indicated that the transcript levels of seven out of the sixteen key enzyme genes in the salvianolic acid biosynthesis pathway were significantly increased, including *SmC4H1*, *SmC4H2*, *Sm4CL1*, *Sm4CL3*, *SmTAT1*, *SmTAT2*, and *SmHPPR1* (Figure 6). Additionally, among the twenty key enzyme genes in the tanshinone biosynthesis pathway, the transcript levels of sixteen genes were significantly increased, including *SmAACT1*, *SmHMGS*, *SmHMGR1*, *SmMK*, *SmMDC* of the MVA pathway, *SmDXS1*, *SmDXR*, *SmMCT*, *SmMDS*, *SmHDS1*, *SmHDR* of the MEP pathway, *SmIDI*, *SmFPPS*, *SmGPPS*, *SmCPS1*, and *SmCYP76AH1* of the downstream pathway (Figure 7). The upregulation of biosynthetic pathway genes correlated with the elevated levels of salvianolic acid and tanshinones observed in the transgenic *Salvia miltiorrhiza* root lines in this study. The above results indicate that *AtMYB2* has a promoting effect on the biosynthesis of both salvianolic acid and tanshinones.

### 2.7. Transcriptomic Analysis of AtMYB2 Transgenic S. miltiorrhiza

To delve deeper into the molecular mechanisms behind the heightened salt tolerance observed in *AtMYB2* transgenic *S. miltiorrhiza*, we conducted a comparative analysis of the global expression profiles between *AtMYB2* transgenic lines and the pCAMBIA1301 lines (Appendix A, Figure 8A). In the *AtMYB2* transgenic lines, compared to the pCAMBIA1301, a total of 418 differentially expressed genes (DEGs) were identified, comprising 202 upregulated genes and 216 downregulated genes (Appendix A). On the basis of the KEGG annotations of DEGs, most genes were involved in proteasome, hormone signal transduction, the MAPK signaling pathway, and phenylpropanoid biosynthesis (Figure 8B). The results suggest that *AtMYB2* can improve the ability of plants to cope with salt stress and the synthesis of secondary metabolites by promoting gene expression at the transcriptional level.

## 3. Discussion

### 3.1. AtMYB2 Promotes Tolerance to Salt Stress by Improving Antioxidant System Function

Gene expression in higher plants is commonly regulated through the interplay of specific transcription factors and cis-acting elements. Among these, the MYB2 protein stands out, playing a crucial role in regulating processes like secondary metabolism [26,27], the meristem and pollen development [28,29], and response to abiotic stress [30]. Many experiments have been conducted to study *Arabidopsis MYB2*, a typical R2R3-MYB gene, and have found that it is associated with improved plant resistance to hormone and environmental stress. Urao et al. discovered that *AtMYB2* expression increased in response to water stress, high-salt conditions, and ABA treatment at the transcriptional level, and serves as a crucial transcriptional activator in regulating gene expression in response to ABA under drought and salt stress [21,31,32]. Using domain mapping, a calmodulin (CaM) binding domain was identified in AtMYB2, and the specific CaM isoform GmCaM4 was found to enhance the DNA-binding activity of AtMYB2. Overexpressing *GmCaM4* in *Arabidopsis* enhances the transcription rate of AtMYB2-regulated genes. This mechanism boosts salt tolerance by promoting the accumulation of proline [33]. When subjected to salt or drought stress, AtMYB2 suppresses the formation of axillary meristems in *Arabidopsis* by downregulating the expression of the RAX1 (regulator of axillary meristem1) gene. This adaptation enables plants to undergo a shortened vegetative developmental stage in response to environmental stresses [28]. Choline monooxygenase (CMO) is an enzyme located in the chloroplast stroma that is responsible for catalyzing the initial step of glycine betaine synthesis in plants. AtMYB2 has the ability to directly interact with the CMO promoter, thereby enhancing plant adaptation to salinity by promoting the accumulation of glycine betaine [34]. In the present study, *AtMYB2* transgenic plants showed a more resistant phenotype under high-salinity conditions compared to the wild-type plants by improving the antioxidant system’s function under oxidative stress.

Salt stress disrupted intracellular ROS homoeostasis, inducing excessive accumulation of ROS and causing oxidative stress to plants. Hence, maintaining low levels of ROS production and bolstering robust ROS-scavenging capabilities are crucial during salt stress. At this stage, the activity of antioxidant enzymes in plants increases to mitigate the damage caused by salt stress [35,36,37]. For instance, overexpression of *Tabzr1-1D* enhances ROS scavenging capacity, thus improving the salt stress resistance in wheat [38]. *AtMYB49* can function to alleviate oxidative damage induced by salinity by upregulating the expression of certain genes encoding POD in transgenic *Arabidopsis* such as *Prx15, Prx24, Prx26, Prx32, Prx53,* and *Prx55* [39]. Meanwhile, *DcWRKY3* positively contributes to the response to salt stress by enhancing the ability to scavenge ROS and by maintaining osmotic pressure balance in plants [40]. Our results indicated that *AtMYB2* transgenic *S. miltiorrhiza* plants exhibited significantly reduced ROS (O_2_^−^ and H_2_O_2_) accumulation compared to WT plants under salt stress conditions (Figure 3), with higher enzymatic scavenger (SOD, POD, and CAT) activities (Figure 4). The NBT and DAB staining also supported the results of the O_2_^−^ and H_2_O_2_ content assay (Figure 3A,B). Meanwhile, compared with the WT, there were lower EL and MDA levels, accompanied by higher total chlorophyll content, in *AtMYB2* transgenic *S. miltiorrhiza* (Figure 4). The aforementioned findings suggest that AtMYB2 boosts the activity of ROS-scavenging enzymes while reducing damage to the lipid structure of plant cell membranes under salt stress, thereby promoting tolerance to salt stress in *S. miltiorrhiza* by improving antioxidant system function. This once again demonstrated the important role of *AtMYB2* in plant response to salt stress, which involves responding to the damage caused by salt stress through various different pathways.

It can be seen that *AtMYB2* plays an important role in plant response to high-salinity environmental stress. Consequently, studying and using it well is conducive to the development, protection, and utilization of plants in harsh environments. For *Salvia miltiorrhiza*, *AtMYB2* is a good candidate gene for improving salt tolerance, which has positive significance for molecular breeding to improve the quality of *Salvia miltiorrhiza*.

### 3.2. AtMYB2 Promotes Tanshinones and Salvianolic Acid Content by Upregulating the Expression of Biosynthesis Genes

Secondary metabolites often accumulate in specific tissues [41]. For instance, artemisinin biosynthesis and accumulation occur in glandular trichomes, while gossypol primarily accumulates in pigment glands in cotton [42]. In the case of *S. miltiorrhiza*, roots serve as the main site for the accumulation of tanshinones and phenolic acids. In *Arabidopsis*, a number of genes have been proven to regulate the content of tanshinones and salvianolic acid. *AtMYC2* positively contributes to the accumulation of both tanshinones and phenolic acids. In the most favorable transgenic line, *AtMYC2-3*, the tanshinone yield reached 14.06 mg/g dw, marking a remarkable 5.45-fold increase compared to the control line. Similarly, phenolic acid production was enhanced to 95.9 mg/g dw, representing a significant 3.3-fold increase compared to the control [43]. Moreover, ectopic expression of *AtWRKY18*, *AtWRKY40*, and *AtMYC2* resulted in increased production of abietane diterpenes, such as carnosic acid and salvipisone, in hairy roots of *Salvia sclarea*. These transcription factors were capable of coordinating the regulation of several genes involved in the MEP-derived terpene pathway [44]. In addition, *AtMYB75* (*AtPAP1*) [19] and *AtMYB12* [20] are both positive regulators of phenolic acid biosynthesis. In this study, we demonstrated that *AtMYB2* can simultaneously promote the content of tanshinones and phenolic acid in both hairy roots and plants by upregulating the expression of biosynthesis genes.

TAT (tyrosine aminotransferase) is the initial key enzyme in the tyrosine-derived pathway, playing a crucial role in the development and metabolic processes of plants [45]. According to the results of qRT-PCR, the expression levels of *SmTAT1* in *AtMYB2-L6* transgenic hairy roots (Appendix A) and *OE*-*AtMYB2-L7* transgenic plants (Figure 6) reached 3.19 times and 3.10 times that of WT, respectively. It was reported that the overexpression of C4H triggers the activation of two parallel pathways for RA biosynthesis, resulting in the accumulation of a variety of downstream metabolites [46]. In this study, the expression levels of *SmC4H1* and *SmC4H2* were significantly increased in *AtMYB2* transgenic plants (Figure 6). As the key enzyme genes in the phenolic acid synthetic pathway, the increase in the expression of these genes plays an important positive role in the accumulation of phenolic acid content. Actually, the SalB content increased by 38.5% (40.5 mg/g dw) in *OE-AtMYB2-L5* transgenic plants when compared with that in the WT (Figure 5H).

In recent years, genes such as *SmGGPPS*, *SmDXR*, *SmCPS1*, *SmKSL1,* and *SmCYP76AH1,* which play important roles in tanshinone metabolism pathway, have been characterized from *S. miltiorrhiza* [47]. It has been reported that *SmERF73* can coordinately regulate the transcriptional expression of seven key enzyme-encoding genes, including *CPS1*; *KSL*; *CYP76AH1*; *CYP76AH3*; *CYP76AK1*; and the upstream MEP pathway genes *DXR1* and *DXS2*, resulting in increased accumulations of tanshinone compounds [48]. In the present study, the expression of 20 genes in both the MEP and MVA pathways was enhanced in *AtMYB2* transgenic plants, especially the *SmHMGS*, *SmMDC,* and *SmGXS1* genes (Figure 7), suggesting that *AtMYB2* may modulate the metabolic flux by regulating the transcript abundance of specific genes, and this regulation led to notable increases in tanshinone concentration. In the current investigation, the augmentation of *AtMYB2* expression yielded substantial enhancements in the concentrations of three tanshinones, including TI, TIIA, and TT (Figure 5H). The augmentations underscored the pivotal regulatory role exerted by *AtMYB2* in the biosynthetic pathways of the bioactive molecules in *Salvia miltiorrhiza*.

The significant augmentation observed in the concentrations of both salvianolic acid and tanshinones underscores the efficacy of *AtMYB2* overexpression as a potent strategy for augmenting the accumulation of secondary metabolites, thus bolstering the pharmacological potential of the plant. These findings hold immense promise for the biotechnological manipulation of medicinal plant species and have positive significance for molecular breeding of *S. miltiorrhiza* with high component content. It is important to note that, while our study sheds light on the regulatory role of *AtMYB2* in salvianolic acid and tanshinone biosynthesis, ongoing research endeavors in this domain continue to unveil new facets of plant metabolic engineering, refining our understanding and offering novel avenues for optimizing the production of valuable bioactive compounds.

## 4. Materials and Methods

### 4.1. Plant Materials and Vector Construction

*S. miltiorrhiza* seedlings were cultivated on solid Murashige and Skoog (MS) medium for 25–30 days, then transplanted into flower pots (8.4 cm diameter) containing a sterile mixture of nutritive soil and vermiculite at a 3:1 ratio. The transplanted seedlings were grown in a controlled environment chamber at 25 ± 2 °C under a 16 h light/8 h dark photoperiod for 30 days. The full-length ORF of the *AtMYB2* gene (NCBI reference sequence AT2G47190) was amplified using gene-specific primers (*AtMYB2-F/R*) and subsequently cloned into the binary vector pCAMBIA1301 to generate the recombinant plasmid *pCAMBIA1301-AtMYB2* (Appendix A). The empty vector and the recombinant AtMYB2 construct were individually transformed into Agrobacterium for plant transformation experiments. The sequences of all primers used in this study are provided in Appendix A.

### 4.2. Genetic Transformation and Molecular Characterization

Transgenic hairy roots of plants were generated via an improved *Agrobacterium*-mediated leaf disk transformation method [49,50] and identified by PCR using primer pairs specific to the 35S promoter (*35S-F*) and *AtMYB2* (*AtMYB2-R*), the rolB gene (*rolB-F/R*), and the hygromycin resistance gene (*Hyg-F/R*) (Appendix A). Positive transgenic hairy root and plant lines confirmed by PCR were subsequently used for further analyses, including phenotypic observation, metabolite profiling, and stress treatment experiments.

### 4.3. RNA Isolation and Detection of Gene Transcription Level

Total RNA was isolated from both WT and transgenic hairy roots/plants following the manufacturer’s protocol [20,50]. The relative expression levels of genes were analyzed relative to *SmActin* using the 2^−ΔΔCT^ calculation method [51,52]. The quantitative primers employed in this study are listed in Appendix A.

### 4.4. Stress Treatment

For the short-term salt treatment, leaf discs with a diameter of 10 mm were extracted from the third-youngest leaves using a hole punch. Subsequently, these leaf discs were promptly transferred to Petri dishes with a diameter of 6 cm, each containing solutions with NaCl concentrations of 0 mM, 100 mM, and 250 mM. Phenotypic changes were observed and documented on the fourth day after treatment.

In the case of the long-term salt treatment, the experimental group underwent irrigation with a 250 mM NaCl solution, and the WT plants irrigated with water (an equivalent volume of 1 L) were used as a control. This irrigation process occurred once every 3 days until the seedlings succumbed on the fifteenth day.

On day 37, plants experienced a 7-day stress period with a 250 mM NaCl solution. At this juncture, physiological indicators were measured on the third-youngest leaf of at least three randomly selected plants per treatment. We used a completely random group design, encompassing three biological replicates.

### 4.5. Determining the Physiological Indices of Leaves and the Active Ingredients in Roots

Physiological parameters, including MDA, SOD, POD, CAT, EL, and chlorophyll content of the transgenic *S. miltiorrhiza* plants and WT plants, were measured according to previously published methods [51]. In the present study, accumulation of O^2−^ and H_2_O_2_ was observed using NBT and DAB staining, respectively [52]. The O^2−^ production rate and H_2_O_2_ content were quantified using a reagent kit (Beijing Solarbio Science and Technology, Beijing, China).

To assess the impact of *AtMYB2* on the accumulation of active ingredients in transgenic hairy roots and plants, we utilized hairy roots cultivated for 60 days and *S. miltiorrhiza* roots harvested after a 5-month growth period in the experimental field for content determination of tanshinones and salvianolic acids. The above materials were treated with authentic standards [20] and quantified using the EClassical 3100 HPLC system (Elite, Dalian, China). For every experiment, three replicates were conducted for each line, with each replicate consisting of three individual plants.

### 4.6. Transcriptomic Analysis of AtMYB2 Transgenic S. miltiorrhiza

Transcriptomic analysis was performed to study the molecular mechanism of salt stress resistance of *AtMYB2* in transgenic *S. miltorrhiza*. Empty vector control pCAMBIA1301 and pCAMBIA1301-AtMYB2 transgenic line 5 were selected for transcriptomic analysis according to previously published methods [20].

### 4.7. Statistical AnalysisOK

The error bars indicate the SD values (n = 3). * *p* < 0.05, ** *p* < 0.01.

## Figures and Tables

**Figure 1 ijms-25-04111-f001:**
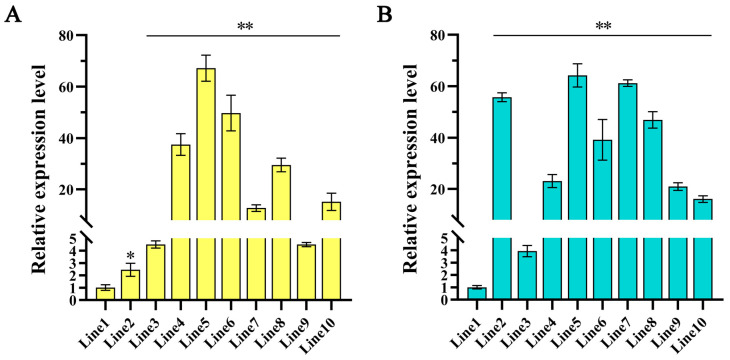
Expression pattern of *AtMYB2* in *S. miltiorrhiza* transgenic hairy roots (**A**) and *S. miltiorrhiza* transgenic plants (**B**). * *p* < 0.05, ** *p* < 0.01.

**Figure 2 ijms-25-04111-f002:**
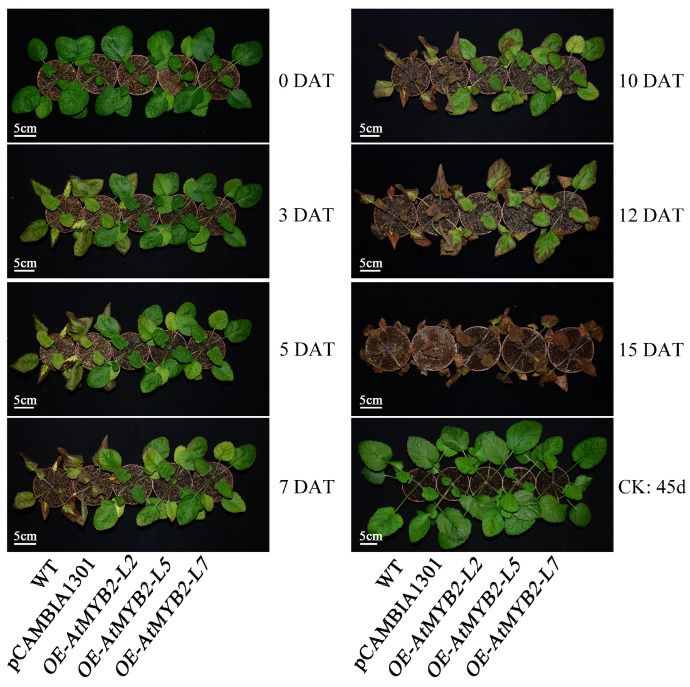
Morphological changes under salt stress. Scale bar = 5 cm; CK, control group; DAT, days after treatment; WT, wild type; pCAMBIA1301, empty vector control pCAMBIA1301 line; *OE-AtMYB2-L2/L5/L7*, *AtMYB2* transgenic lines 2/5/7.

**Figure 3 ijms-25-04111-f003:**
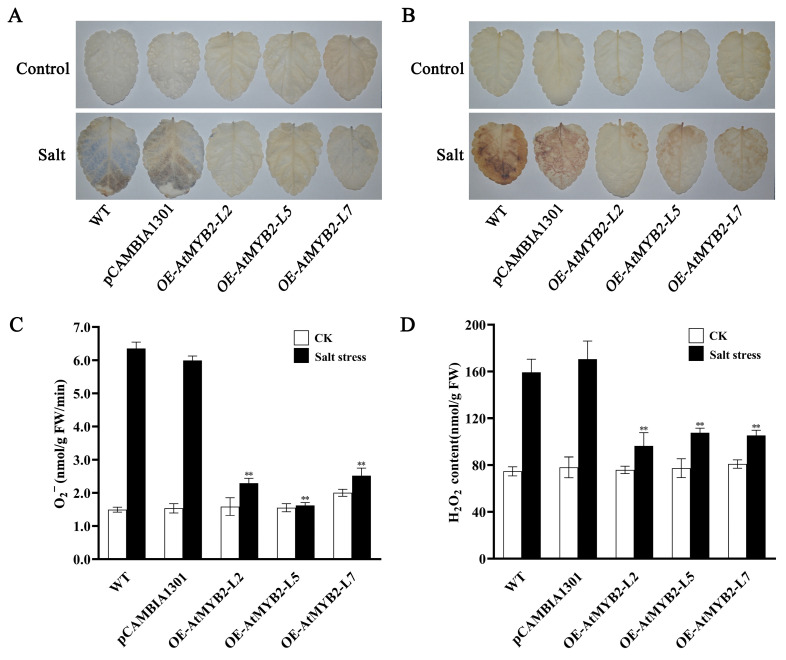
Accumulation of O^2−^ (**A**,**C**) and H_2_O_2_ (**B**,**D**) in all *S. miltiorrhiza* lines under salt stress. The localization and accumulation of O^2−^ (**A**) and H_2_O_2_ (**B**) were visualized by blue and brown staining. Control, plants grown under normal conditions; Salt, plants subjected to salt stress treatment for 7 days. ** *p* < 0.01.

**Figure 4 ijms-25-04111-f004:**
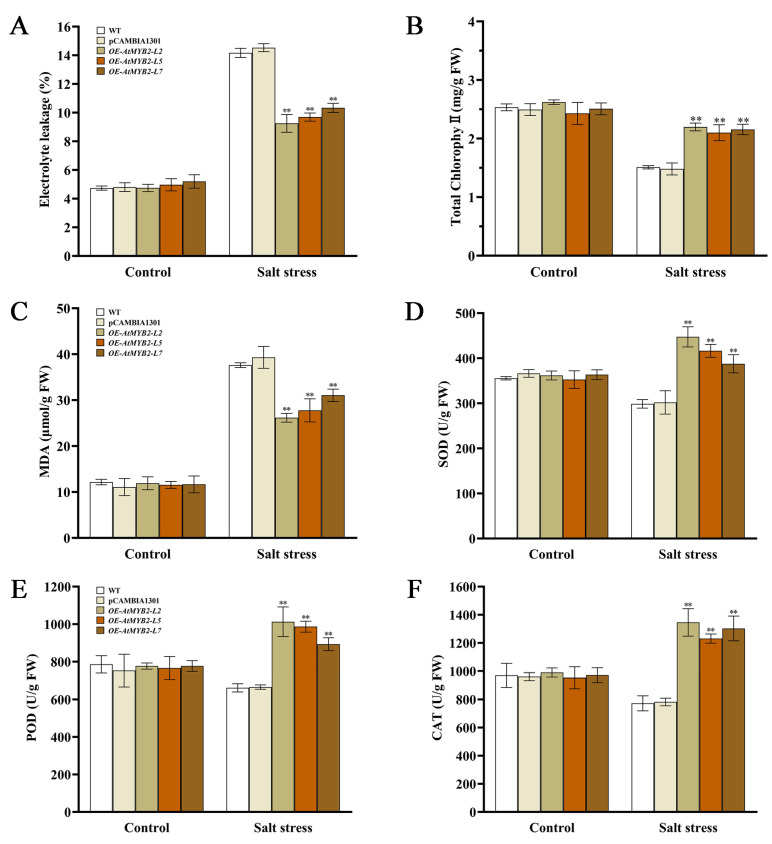
*AtMYB2* improves antioxidant capacity under salt stress. Content of EL (**A**), total chlorophyll (**B**), and MDA (**C**) on 7 DAT; activities of SOD (**D**), POD (**E**), and CAT (**F**) in all *S. miltiorrhiza* lines on 7 DAT. ** *p* < 0.01.

**Figure 5 ijms-25-04111-f005:**
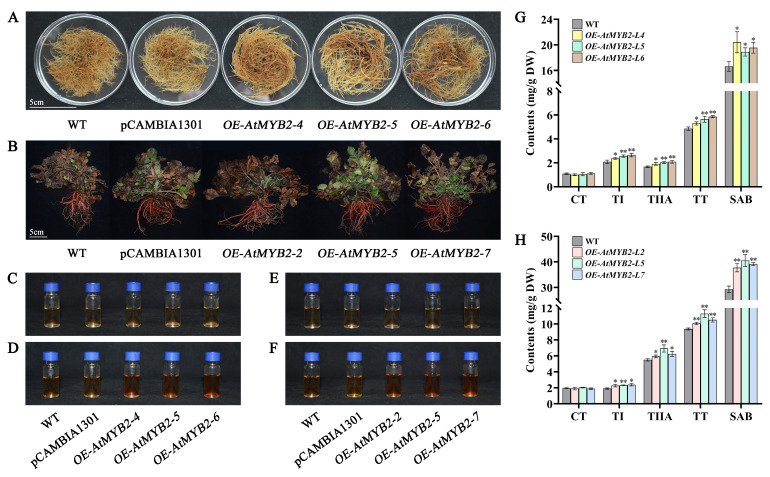
*AtMYB2* promotes the accumulation of tanshinones and salvianolic acid. The phenotype of harvested *S. miltiorrhiza* hairy roots and *S. miltiorrhiza* plants, scale bar = 5 cm (**A**,**B**); the salvianolic acid B (**C**) and tanshinones (**D**) extracts from the *AtMYB2* transgenic hairy root lines; the salvianolic acid B (**E**) and tanshinones (**F**) extracts from the *AtMYB2* transgenic plant lines; content of secondary metabolites in transgenic hairy roots (**G**) and plants (**H**). * *p* < 0.05, ** *p* < 0.01.

**Figure 6 ijms-25-04111-f006:**
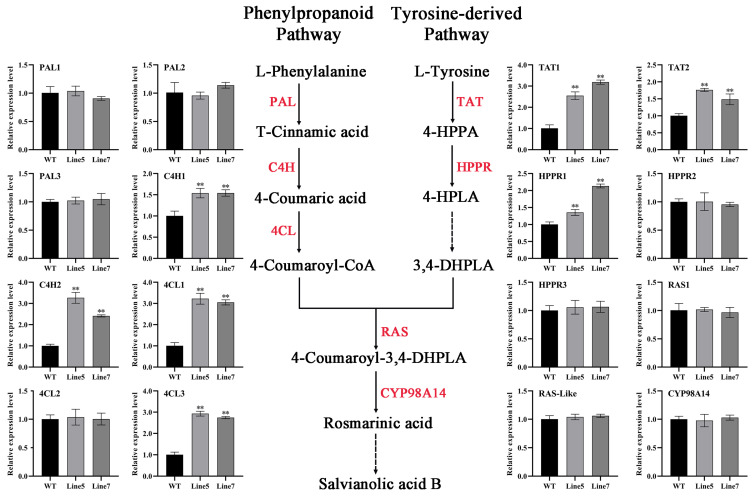
Transcription levels of genes involved in the salvianolic acid biosynthesis pathway in transgenic *Salvia miltiorrhiza* plants. ** *p* < 0.01.

**Figure 7 ijms-25-04111-f007:**
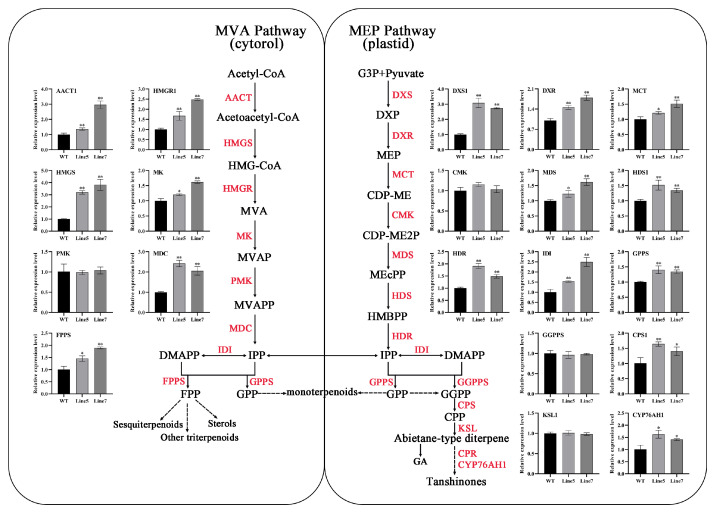
Transcription levels of genes involved in the tanshinone biosynthesis pathway in transgenic *Salvia miltiorrhiza* plants. MVA pathway, the mevalonate pathway in the cytosol; MEP pathway, 2-C-methyl-d-erythritol-4-phosphate pathway in the plastids. * *p* < 0.05, ** *p* < 0.01.

**Figure 8 ijms-25-04111-f008:**
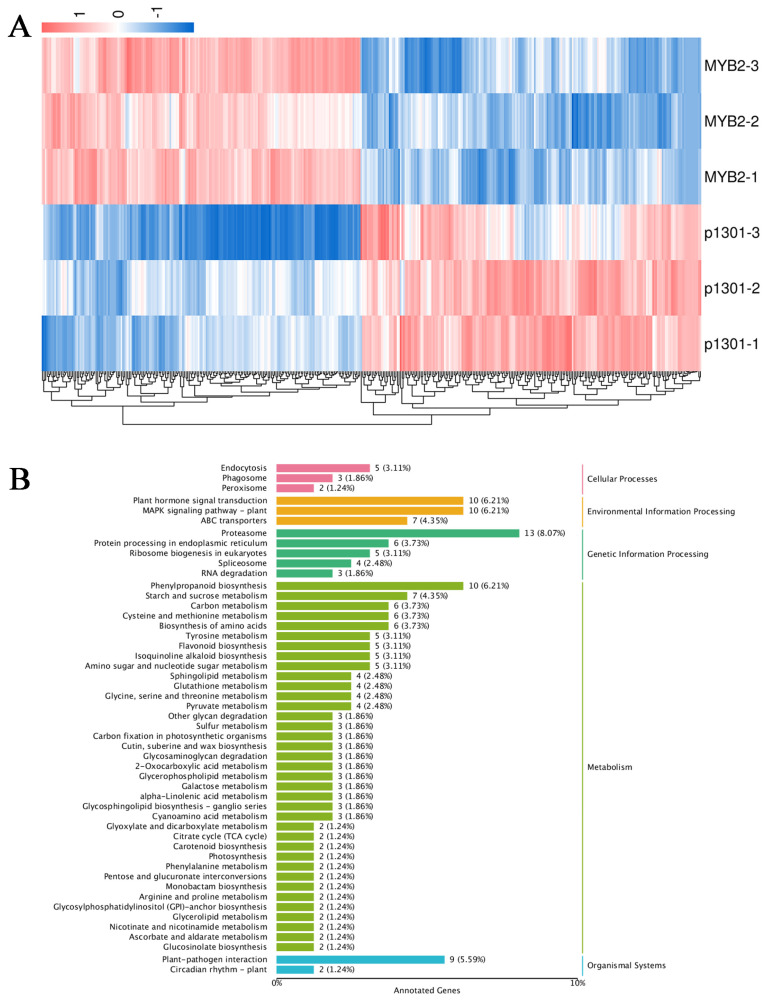
Hierarchical clustering of DEGs of *AtMYB2* transgenic lines versus those of pCAMBIA1301 plants (**A**) and KEGG classification of DEGs of *AtMYB2* transgenic lines versus those of pCAMBIA1301 plants (**B**).

## Data Availability

The original data of this present study are available from the corresponding authors.

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
