# Peer review of "Overexpression of *AtMYB2* Promotes Tolerance to Salt Stress and Accumulations of Tanshinones and Phenolic Acid in *Salvia miltiorrhiza"

_ijms, 2024, doi:10.3390/ijms25074111_

Round 1

Reviewer 1 Report

Comments and Suggestions for Authors

Manuscript "Overexpression of AtMYB2 promotes tolerance to salt stress and accumulation of tanshinones and phenolic acid in Salvia miltiorrhiza" by authors

Tianyu Li, Shuangshuang Zhang, Yidan Li, Hongpeng Wang, Wenqin Song, Chengbin Chen contains a lot of duplicate information, ranging from photographic materials, methods, genes and other information, the differences affect only the part of the article in which expression analysis is carried out. The data presented raise doubts and I recommend that the authors rewrite the article, citing not their previously published work:

Li, T.; Zhang, S.; Li, Y.; Zhang, L.; Song, W.; Chen, C.; Ruan, W. Simultaneous Promotion of Salt Tolerance and Phenolic Acid Biosynthesis in Salvia miltiorrhiza via Overexpression of Arabidopsis MYB12. Int. J. Mol. Sci. 2023, 24, 15506. https://doi.org/10.3390/ijms242115506

All duplicate data should be removed and provided only in the discussion.

The article cannot be published in its present form. If the authors want to use images of plants in the future, they should provide the dimension (ruler) and not use plants that have already been used, since the availability of the sample is questionable. The same issue applies to sampling in expression analysis.

Author Response

Response: Thank you very much for the comments from you about our manuscript submitted to International Journal of Molecular Sciences. The manuscript entitled “Overexpression of AtMYB2 promotes tolerance to salt stress and accumulations of tanshinones and phenolic acid in Salvia miltiorrhiza” (ijms-2894996) has been revised in detail according to the comments. The corrections have been done and indicated in red color in the revised manuscript. Thanks again for your helpful suggestion. All the comments have been incorporated into the revised manuscript. The responses (in BOLD type) to the comments are stated below:

Reviewer 2 Report

Comments and Suggestions for Authors

Reviewer comment

Although the manuscript is worthy of being published in this high esteem journal due to the vital information embedded in it; still the manuscript has some drawbacks regarding technical issues that I noticed in the following comments;

Overall, I recommend this article as a “major revision

1. Title: Please make correction “accumulation”

2. Abstract is the mirror of the article it should be figure out whole story of article. Generally, abstracts should give the highlights of the findings obtained, but focus on making a coherent story, rather than simply a listing of results.

           Please make clear what tissue you used in this study and the proper reason behind using those tissues regarding the mention abiotic stress (Salinity stress)

3. In the introduction, the authors are asked to better describe tanshinones and phenolic acid biosynthesis role during salinity and their link with the cardiovascular and cerebrovascular diseases disease.

4. To enable efficient reproducibility of the results by readers, it's crucial for authors to provide the specific details on the data availability, such as the specific gene primers as well as the RNA-seq data.

5. In the results 2.7, Since the Transcriptome data was used, the basic information of Transcriptome sequencing should be provided, such as the total amount of sequencing data, the average reading length, the longest sequence, and the situation of differential genes.
6. It would be more informative if this paper could include a discussion of the broader applications of the study's findings for crop improvement and identify potential areas for further investigation.
7. Some sentences in the introduction and discussion section need to be polished by a native speaker

8. The experimental design was not explained; i.e. did the authors employ, for example, a complete randomized block design, etc.

9. In materials and methods section, for quantitative RT-PCR it should be presented what Ct values are observed for reference gene, which is ACTIN, as it sometimes is not stable under severe stress or among different tissues. Moreover, it should be described if DNase treatment was applied before RT.

10. In the Materials & Methods section, some references should be added since several protocols are from the reported methods or experimental procedures i.e RNA sequencing, qPCR

11.Some Figure looks blur. You should change it

Comments on the Quality of English Language

English expression is good however few area need to revised.

Author Response

Response: Thank you very much for the comments from you about our manuscript submitted to International Journal of Molecular Sciences. The manuscript entitled “Overexpression of AtMYB2 promotes tolerance to salt stress and accumulations of tanshinones and phenolic acid in Salvia miltiorrhiza” (ijms-2894996) has been revised in detail according to the comments. The corrections have been done and indicated in red color in the revised manuscript. Thanks again for your helpful suggestion. All the comments have been incorporated into the revised manuscript. The responses (in BOLD type) to the comments are stated below:

  1. Title: Please make correction “accumulation”.

Response: Thank you for the comments. The corrections have been done and indicated in red color in the revised manuscript.

  1. Abstract is the mirror of the article it should be figure out whole story of article. Generally, abstracts should give the highlights of the findings obtained, but focus on making a coherent story, rather than simply a listing of results.

Response: Thank you for the useful comments. The abstract has been revised according to your suggestions and indicated in red color in the revised manuscript.

  1. Please make clear what tissue you used in this study and the proper reason behind using those tissues regarding the mention abiotic stress (Salinity stress)

Response: Thanks for your comments. For materials related to salinity stress in this study, leaf tissue (7 days after treatment) was used. This appears by reference in the original text, which may not be clear, and has been amended.

  1. In the introduction, the authors are asked to better describe tanshinones and phenolic acid biosynthesis role during salinity and their link with the cardiovascular and cerebrovascular diseases disease.

Response: Thanks for your comments. Background on the association of tanshinones and salvianolic acid with cardiovascular and cerebrovascular diseases has been added to the third paragraph of the “introduction” section. Additional content is marked in red color in the revised manuscript.

  1. To enable efficient reproducibility of the results by readers, it's crucial for authors to provide the specific details on the data availability, such as the specific gene primers as well as the RNA-seq data.

Response: Thank you for the valuable comments. We have supplemented the relevant transcriptome data and primer information in the supplementary material of the article.Figure S5 and table S2.

  1. In the results 2.7, Since the Transcriptome data was used, the basic information of Transcriptome sequencing should be provided, such as the total amount of sequencing data, the average reading length, the longest sequence, and the situation of differential genes.

Response: Thank you for the comments. The basic information of transcriptome sequencing has been added in the supplementary materials of the article (Table S1).

  1. It would be more informative if this paper could include a discussion of the broader applications of the study's findings for crop improvement and identify potential areas for further investigation.

Response: Thank you for the useful comments. Relevant content has been supplemented to the discussion section of the article and indicated in red color in the revised manuscript.

  1. Some sentences in the introduction and discussion section need to be polished by a native speaker.

Response: Thank you for the comments. The introduction and discussion have been polished in English.

  1. The experimental design was not explained; i.e. did the authors employ, for example, a complete randomized block design, etc.

Response: Thank you for the useful comments. The experimental designs in this paper are all completely randomized block design and have been indicated in red color in the revised manuscript.

  1. In materials and methods section, for quantitative RT-PCR it should be presented what Ct values are observed for reference gene, which is ACTIN, as it sometimes is not stable under severe stress or among different tissues. Moreover, it should be described if DNase treatment was applied before RT.

Response: Thank you for the comments. SmActin is the most commonly used reference gene in Salvia miltiorrhiza. Smactin can stably expressed in different tissues of Salvia Miltiorrhiza according to our previous experiments. There have been no reports of its instability so far. RNA reverse transcription was indeed previously processed with DNase, but it was not explicitly noted in the material method because of the kit used.

  1. In the Materials & Methods section, some references should be added since several protocols are from the reported methods or experimental procedures i.e RNA sequencing, qPCR.

Response: Thank you for the comments. The corresponding references have been added to the Materials Methods section where required.

  1. Some Figure looks blur. You should change it English expression is good however few area need to revised.

Response: Thank you for the useful comments. The clarity of all figures was checked and optimized. The full text of the English expression has been polished and modified.

Round 2

Reviewer 1 Report

Comments and Suggestions for Authors

Article "Overexpression of AtMYB2 promotes tolerance to salt stress and accumulations of tanshinones and phenolic acid in Salvia

miltiorrhiza" by Tianyu Li, Shuangshuang Zhang, Yidan Li, Lipeng Zhang, Wenqin Song and Chengbin Chen submitted for review has been supplemented and changed, but I did not see answers to the questions and ask for them (All the comments have been incorporated into the revised manuscript . The responses (in BOLD type) to the comments are stated below:...?).Especially in terms of photos and links. I also recommend enlarging the image Figure 8.

Author Response

Thank you for your comments. Based on the suggestions you provided, we have made the following modifications to the article:

  1. Creating clearer images and adding them to the article, adjusting their sizes appropriately.
  2. Rearranging and analyzing the experimental data, presenting it in a clearer manner in the article.
  3. Adding a scale bar to the plant images.
  4. Reducing the descriptions referencing the previously published article (ijms242115506).